# OpenReview forum: "Large Language Model Benchmarks Do Not Test Reliability"
_NeurIPS.cc/2024/Workshop/SafeGenAi — SafeGenAi Poster_

### Official Review · Reviewer_xkPS · 2024-10-08
**Review for Large Language Model Benchmarks Do Not Test Reliability**

**Rating:** 7
**Confidence:** 5

**Review:**

This paper proposes construction of platinium benchmarks; benchmarks on which a reliable and performant model should achieve 100% accuracy. The paper is tackling an important problem and its motivation is clear. A few comments below:
1. The authors should clarify whether there was any selection criteria in relabeling subsets of benchmarks.
2. The results of evaluation of Platinium benchmarks on LLMs is not clear from Table 4, not even in Appendix results. Please summarise how did the cleaned benchmarks perform versus the original ones. For eg, what does -81% difference mean?
3. Some examples categorised as Ambiguous are due to prompt brittleness. Have the authors tackled this issue in their work?
4. Please fix errors like "Table XXX (row X)"

---

### Official Review · Reviewer_nBMH · 2024-10-09
**Review of "Large Language Model Benchmarks Do Not Test Reliability"**

**Rating:** 6
**Confidence:** 4

**Review:**

## Review
The paper addresses a critical and timely issue in the field of artificial intelligence: while LLMs have shown remarkable capabilities, their reliability remains questionable, particularly in real-world applications where consistent accuracy is paramount.

## Quality
The quality of the research is commendable. The authors pinpoint a significant oversight in current benchmarking practices concerning the assessment of model reliability. They also Introduced platinum benchmarks is a practical approach to push models towards achieving 100% accuracy on tasks they should reliably handle.
They conduct experiments using ten existing benchmarks across various domains, meticulously cleaning them to eliminate ambiguities and errors. And the paper provides detailed methodologies for cleaning the datasets and evaluating the models, enhancing reproducibility.

## Clarity
The paper is well-written and organized, making it accessible to a broad audience. The notion of "platinum benchmarks" is introduced and elaborated upon clearly. Figures and tables are used to demonstrate the issues with current benchmarks and the impact of their cleaning process. The progression from identifying the problem to proposing and evaluating a solution is coherent.

## Originality
On the novel perspective, they shifting the focus from expanding capabilities to enhancing reliability is a fresh angle in LLM research.
Also, the concept of platinum benchmarks is a novel idea that challenges the current lifecycle of benchmarks in AI.

## Significance
The significance of the paper is substantial. Reliable LLMs are crucial for deployment in sensitive areas like healthcare, finance, and legal services.

## Cons
- The scope of the benchmark is limited. Although ten benchmarks are cleaned, they didn't cover all aspects of LLM capabilities, potentially limiting the generalizability of the findings.
- The paper could delve deeper into why models fail on these simple tasks, possibly exploring underlying causes or common patterns in the errors.
- Scalability: Cleaning benchmarks is labor-intensive, and scaling this approach to the vast array of existing and future benchmarks may be challenging.

## Conclusion
The paper makes a significant contribution by highlighting a crucial yet overlooked aspect of LLM evaluation—reliability on tasks where models are expected to perform perfectly. The introduction of platinum benchmarks is a practical and impactful proposal that could reshape how the AI community assesses model performance. While there are areas for further exploration, such as understanding the root causes of model failures and expanding the range of cleaned benchmarks, the work lays a good foundation for future research aimed at closing the gap between a model's capabilities and its reliability.

---

### Official Review · Reviewer_JXNu · 2024-10-10
**Not sure if the results have strong implications**

**Rating:** 3
**Confidence:** 4

**Review:**

The paper shows the popular benchmarks like MMLU and GSM8K have mislabeled examples. The paper corrects these issues and then tests modern LLMs again on these corrected samples. The performance on these samples is not 100%. Using this evidence, the paper argues that the LLMs have not "beaten" the benchmarks in the sense that the error is not simply due to labeling issues. The paper urges that the LLMs should still be tested on old (corrected) benchmarks.

Unfortunately, while the insight it interesting, there seems to be an overselling aspect to how it is presented. I do not think the additional performance gain achieved on these benchmarks (e.g., additional 5% on GSM8K) adds more "reliability" to the model. The term reliability seems to imply that the model is more robust (e.g., to prompt rephrasing, simple mistakes like spelling errors) and trustworthy (e.g., less biased, able to express its uncertainty well). However, neither of these aspects is studied in the paper. Moreover, the paper does not show that the additional 5% performance would lead the model to generalize better to real world math problems. So the usage of term "reliability" is questionable.

I still think the insight, that the existing benchmarks have mislabeled examples, and the model can still make mistakes on corrected versions to them, is interesting. However, the error rates (in Table 3) are relatively small and not clear if they show any profound underlying mechanism.